# Effect of Vesicle Size on the Cytolysis of Cell-Penetrating Peptides (CPPs)

**DOI:** 10.3390/ijms21197405

**Published:** 2020-10-07

**Authors:** Kazutami Sakamoto, Takeshi Kitano, Haruka Kuwahara, Megumi Tedani, Kenichi Aburai, Shiroh Futaki, Masahiko Abe, Hideki Sakai, Hiroyasu Ohtaka, Yuji Yamashita

**Affiliations:** 1Tokyo University of Science, Noda, Chiba 278-8510, Japan; t-kitano@patent.wpa.co.jp (T.K.); pal0505pal@yahoo.co.jp (H.K.); ken.aburai@gmail.com (K.A.); abemasa@rs.tus.ac.jp (M.A.); hisakai@rs.noda.tus.ac.jp (H.S.); 2Chiba Institute of Science, Choshi, Chiba 288-0025, Japan; m_tedani@takemotokk.co.jp (M.T.); hohtaka@cis.ac.jp (H.O.); yyamashita@cis.ac.jp (Y.Y.); 3Institute for Chemical Research, Kyoto University, Uji, Kyoto 611-0011, Japan; futaki@scl.kyoto-u.ac.jp

**Keywords:** cell-penetrating peptide (CPP), direct permeation (cytolysis), giant unilamellar vesicle (GUV), FITC–octa-arginine (FITC–R8), vesicle size, gel phase, liquid crystal (LC)

## Abstract

A specific series of peptides, called a cell-penetrating peptide (CPP), is known to be free to directly permeate through cell membranes into the cytosol (cytolysis); hence, this CPP would be a potent carrier for a drug delivery system (DDS). Previously, we proposed the mechanism of cytolysis as a temporal and local phase transfer of membrane lipid caused by positive membrane curvature generation. Moreover, we showed how to control the CPP cytolysis. Here, we investigate the phospholipid vesicle’s size effect on CPP cytolysis because this is the most straightforward way to control membrane curvature. Contrary to our expectation, we found that the smaller the vesicle diameter (meaning a higher membrane curvature), the more cytolysis was suppressed. Such controversial findings led us to seek the reason for the unexpected results, and we ended up finding out that the mobility of membrane lipids as a liquid crystal is the key to cytolysis. As a result, we could explain the cause of cytolysis suppression by reducing the vesicle size (because of the restriction of lipid mobility); osmotic pressure reduction to enhance positive curvature generation works as long as the membrane is mobile enough to modulate the local structure. Taking all the revealed vital factors and their effects as a tool, we will further explore how to control CPP cytolysis for developing a DDS system combined with appropriate cargo selection to be tagged with CPPs.

## 1. Introduction

The cell membrane is composed of a lipid bilayer that works as a barrier to control the inflow and outflow of substances. Generally, substances get into or out of cells under the control of membrane proteins such as channels and receptors to maintain viability and homeostasis. However, a specific series of peptides, called a cell-penetrating peptide (CPP), is known to be free to directly permeate through cell membranes into the cytosol (cytolysis); hence, this CPP could be a potent carrier for a drug delivery system (DDS). Therefore, elucidation of the cytolysis mechanism is essential to understanding CPPs’ transmembrane penetration [1,2,3,4]. Wender et al. revealed the necessary structural requirements to be a CPP, which has plural strong cationic moiety, such as guanidium groups that are close together. They also found that the peptide backbone is not indispensable and showed that peptoid or even polyamine could be a potential CPP [4]. These data suggest that cytolysis is not a specific molecular interaction, like enzymatic reaction or ligand and receptor interactions, between CPPs and the membrane lipid. Additionally, we have found cytolysis, as a structural modification of the lipid bilayer, to be a hydrated lamellar liquid crystal (Lα) [5,6]. Israelachvili and Ninham et al. revealed the general theory that the topology of self-assembled surfactant as liquid crystal (LC) changes structure from Lα thorough bicontinuous (V_1_) to hexagonal (H_1_) crystal by modulating the molecular geometry parameter called the surfactant parameter (SP) [7,8] as shown in Appendix A [9]. Namely, *SP* correlates to the type of liquid crystal (LC) structure of the lipid membrane by changing the cross-sectional area (*a*) at the hydrophilic part of the lipid, as shown by Equation (1).
*SP* = *v*/(*a* × *l*)(1)
where *v* is the hydrophobic part of the volume, and *l* is the hydrophobic chain length [7,8]. Hyde et al. reported the importance of intermediate structures between lamellar (Lα) and hexagonal (H_1_), especially the Mesh_1_ phase, for biological functions. Interestingly, Mesh_1_ has a unique structure of undulated lamellar bilayers, with periodical pores to connect water layers, as shown in Appendix A [10]. 

Based on this theory and noticing the structure of Mesh_1_, we hypothesized cytolysis’ mechanism to be related to the phase transfer of membrane lipid from Lα to Mesh_1_ [5,6,9], as shown in Appendix A. We have reported that oligo-arginine (Rx) as a CPP changed the Lα structure, composed of membrane lipids, under the equilibrium condition based on the *SP* theory. Octa-arginine (R8) reduced the *SP* value of Lα of 1-oleyl-2-hydroxy-phosphatidylcholine (OPC) with water. Namely, by the incremental addition of R8 to the Lα composed of an OPC/water mixture, Lα changed to H_1_ through a V_1_ phase, reflecting the *SP* reduction by Equation (1) [5,6,9]. Additionally, hexa-arginine (R6) made the same *SP* reduction for Lα of di-oleoyl-3-phosphatidylcholine (DOPC) with water. Interestingly, a small amount of R6 increased the interlayer spacing of the Lα lamellar structure attributed to the curved deformation of planar Lα to the undulated lamellar layers, with periodical pores as the Mesh_1_ phase [11]. From these findings, we proposed the mechanism of cytolysis as a temporal and local phase transfer of membrane lipid from Lα to Mesh_1_ [6,9,11], as shown in Appendix A [12]. Moreover, we predicted that we could control CPP cytolysis by changing the membrane curvature.

To show the curvature effect for cytolysis, we choose erythrocyte, which has negative mean curvature. Erythrocyte can flexibly change its shape because of its biconcave disk shape. We found erythrocyte to be resistant to cytolysis because of its negative mean curvature. Under lower osmotic pressure, the cytolysis of CPPs to erythrocyte increased along with the swelling to make the membrane curvature positive [5,6,9,10]. We have recently investigated the key process and factors controlling cytolysis as direct translocation of CPPs through biomembranes [12]. As a result, we found that adsorption and cytolysis are assumed to occur successively when the CPP molecule comes into contact with membrane lipids. Adsorption is quick to saturate, while cytolysis proceeds at a constant rate in time for each adsorbed CPP molecule. By changing the osmotic pressure, we confirmed that the membrane curvature is the key to cytolysis as a local and temporal phase transition from Lα to Mesh_1_. We also found that temperature and lipid composition influence cytolysis through modulating lipid mobility. As such, we further investigated the most critical factor for cytolysis, which is our objective in this report. 

Our strategy is to control membrane curvature by changing the vesicle size based on the curvature (*κ*) and radius (*r*) relationship of Equation (2) as a general mathematical expression [13].
*r* = 1/*κ*(2)

Based on this equation, we have utilized the effect of the vesicle’s size on CPP cytosis because this physical modulation is the most straightforward method. After all, we can keep all the other experimental conditions the same, with only vesicle size as a variable. This study expects to find enhanced cytolysis for smaller vesicles, and we conducted experiments for phospholipid vesicles with diameters from 0.1 to 14 μm. To our surprise, we found that the smaller the vesicle diameter (meaning a higher membrane curvature), the more cytolysis was suppressed. Such controversial findings led us to seek the reason for the unexpected results and we ended up finding out that the flexible mobility of the membrane lipid as a liquid crystal is the key to cytolysis. Namely, the temperature effect for the phase transition of membrane lipid from liquid crystal to a gel state, at which hydrophobic chains stacked to semicrystals, led us to this conclusion. As a result, membrane lipids must be flexible and mobile under the liquid crystal state to change membrane structures through reduced osmotic pressure, leading smaller *SP* values to make Mesh_1_ structures.

## 2. Results

### 2.1. Confirmation of Vesicle Formation

The formation of spherical-shaped vesicles for 1-stearoyl-2-oleoyl-phosphatidylcholine (SOPC) was confirmed by freeze-fractured transmission electron microscope (FF-TEM) or differential interference contrast microscope (DICM) observation, as shown in Figure 1. 

The size distribution of prepared vesicles for egg yolk phosphatidylcholine (EPC) and SOPC were measured by dynamic light scattering (DLS). The result for EPC vesicles is shown in Figure 2. Similar size distribution profiles were also confirmed for SOPC vesicles (DLS profile for SOPC is not shown here).

### 2.2. Calculation of the Geometrical Parameters for Vesicles

The geometric parameters of vesicles were calculated with the average diameter from Figure 2 for each vesicle size, summarized and shown in Table 1 using equations shown in Appendix A [12].

### 2.3. Measurement of Permeation Amount of Cell-Penetrating Peptide (CPP)

#### 2.3.1. Effect of Particle Size of Vesicle on the Cytolysis of FITC–R8 as CPP for EPC Vesicles

The effect of particle size of vesicle on the FITC–R8 cytolysis for EPC vesicles was conducted at 37 °C for 10 min under different osmotic pressure conditions from 14 to 112 mOsm, with the fixed molar ratio of lipid/CPP peptide (L/P) at 1000. We expected to find enhanced cytolysis for smaller vesicles, which have larger curvatures. As shown in Figure 3, the cytolysis amount of CPPs per outer membrane lipid (μmol/mol lipid) decreased when reducing vesicle size throughout the experiments, contrary to our expectation. In particular, CPPs hardly penetrated the vesicles that were smaller than 500 nm. The reduction of osmotic pressure, corresponding to positive curvature generation, enhanced cytolysis as expected except for the vesicles larger than 800 nm, in a good agreement with previous studies. As such, we confirmed our proposed mechanism and the effect of positive curvature by reduced osmotic pressure to enhance cytolysis [6,9]. On the other hand, size reduction opposed and even prevented cytolysis for vesicles under 500 nm. 

We found two different phases of linear relationships between cytolysis and vesicle size over or under the diameter of around 1000 nm. For vesicles larger than 1000 nm, lower osmotic pressure than isotonic condition (hypotonic) showed higher cytolysis. Namely, under hypotonic conditions, it is possible to generate positive curvature to enhance cytolysis. In contrast, a hypertonic state develops negative curvature that cooperatively suppresses cytolysis with size reduction.

The other mode for a vesicle diameter less than 1000 nm is different. All the lines come to convergence at 100 nm, where cytolysis is practically inhibited.

Next, we calculated the average numbers of FITC–R8 molecules per single vesicle by applying geometric parameters from Table 1, which are shown in Figure 4. The linear relationship shown in Figure 4 indicates that the cytolysis of CPP is energy-dependent phenomena. Assuming that the local and temporal pore formation to the Mesh_1_ phase from the Lα phase is the cause of CPP permeation, the depreciation of area difference elasticity (ADE) [14,15,16] or surface energy generated by CPP absorption is the key of this phenomena. Reduced vesicle size correlates to the incremental pressure difference between the inside and outside of the membrane expressed by the Laplace equation (Equation (3)).
Δ*P* = (*P_out_* – *P_in_*) = 2*γ*/*r*(3)
where Δ*P* is the pressure difference between the inside and the outside of the membrane, and *γ* is the surface tension. Under the curved deformation from the planar membrane, there is increased energy as Δ*P* corresponds to the affected membrane surface’s total elastic energy (*F_t_*). *F_t_* can be expressed by Equation (4) [14,15,16]
*F_t_* = *F_b_* + *F_ADE_*(4)
where *F_b_* is the bending energy of the membrane, and *F_ADE_* is the area difference in elasticity, which is associated with the relative stretching of monolayers in the bilayer. Namely, curvature generation requires Δ*P* (or *F_t_*), which corresponds to the increased free energy. This free energy is supplied by increasing intermolecular interaction between hydrophobic chains of membrane lipid as Δ*H*, which sacrifices lipid mobility. As cytolysis occurs under the fluctuation of membrane structure with the prerequisite that the membrane lipid is under a liquid crystal state, molecules can quickly diffuse or flip-flop within the membrane. As such, a reduction of vesicle size suppresses cytolysis through the interference of membrane lipid mobility.

This analysis is a new insight of the cytolysis mechanism to indicate the importance of the mobility of membrane lipids to accommodate *SP* in order to make a temporal and local phase transition to Mesh_1_ under the fluctuation of membrane lipids. Although we expected the curvature increase in size reduction, we found that the mobility of membrane lipids is a far more critical factor for vesicles under 1000 nm. In terms of membrane flexibility, we reported the effect of cholesterol on suppressing cytolysis by reducing the mobility of membrane lipids [12]. Yamazaki et al. also observed the same effect of cholesterol and the impact of chain length in phospholipids on the internalization of CF-R9 [17], even though both of us only discussed the GUV and missed the importance of vesicle size. It is noteworthy to consider that cell size, as an independent self-reproducible existence in life, is usually over 1000 nm, probably because membrane lipids are then free to fluctuate to keep the cell viable [13].

#### 2.3.2. Measurement of CPP Permeation under the Gel Phase

This study examined how membrane lipids’ phase transition affects CPP permeation when the vesicle size is changed. To investigate this, the lipids used should have an appropriate phase transition temperature (Tc) from the Lα phase (liquid crystal) to the gel phase (hydrated solid-state) under the experiment. Egg yolk phosphatidylcholine (EPC) is not suitable for this study, as its T_C_ is under −15 °C. We used 1-stearoyl-2-oleoyl-phosphatidylcholine (SOPC) with Tc 6 °C to investigate the phase effect for CPP cytolysis. CPP cytolysis concerning the vesicle size was almost similar to EPC and SOPC at 30 °C. (data are not shown) Then, we compared the effect of the phase condition of SOPC, i.e., Lα or gel, to cytolysis, as shown in Figure 5. Cytolysis did not occur, irrespective of the vesicle size, at 0 °C in the gel phase. This result indicates that the rigid lipid structure prevents cytolysis even under the Lα phase (liquid crystal state) at 30 °C.

In terms of the temperature effect, the cytolysis profile vs. particle size was almost similar at temperatures over 30 °C, as shown in Figure 5. At 15 °C, closer to the Tc (6 °C), cytolysis declined to some extent, although there was still clear dependence on vesicle size. From this result, it is suggested that the lipid’s phase state is a crucial factor for cytolysis. Visualization of CPP penetration was carried out, as shown in Appendix A. From the image in Appendix A [13], it was confirmed that CPP permeated the vesicles at the LC phase but not at the gel phase. Although it is difficult to judge, vesicles with a small particle diameter in the LC phase seem to have a darker inner layer, corresponding to suppression. A similar example has been reported on this point. Almeida et al. observed that cytolysis is different between large vesicles and small vesicles using antimicrobial peptide (AMP)-type CPPs [18]. Although there was no explanation in their report, we believe this is another independent piece of evidence for our assumption that smaller and more rigid vesicles are resistant to CPP permeation.

### 2.4. Effect of Particle Size of the SOPC Vesicle on the Physicochemical Properties of the Lipid Membrane

#### 2.4.1. Fluorescent Anisotropy Measurement for Lipid Mobility Determination

Fluorescent anisotropy of 1,6-diphenyl-1,3,5-hexatrien (DPH) as a probe dissolved in the membrane lipid represents the mobility of lipid molecules. As shown in Appendix A, the anisotropy of DPH jumped up below Tc at the gel phase, which corresponds to the lipid mobility in reverse. There is an inverse relationship between DPH anisotropy or cytolysis of CPP (FITC–R8) vs. vesicle size at 30 °C, as shown in Appendix A [13]. These results reasonably explain the cytolysis suppression of smaller vesicles, as discussed before, because lipids in the smaller vesicle are less mobile.

#### 2.4.2. Differential Scanning Calorimetry (DSC) Measurement for the Thermal Properties of SOPC Vesicles

Figure 6 shows the DSC peaks of DOPC vesicles for the thawing process from the gel to Lα transition. Phase transition temperatures were almost the same (around 5.6 °C) and there was good concordance with the Tc reported [19], but the shape and area of peaks differed by the vesicle size. There is a pretransition peak only for GUV, which might be attributed to a ripple gel phase transition [20]. The shape of the phase transition peaks to Lα was sharp, with a smaller half-width for GUVs, and the phase transition enthalpy (ΔH) calculated from the area was smaller, as shown in Figure 7. These data suggest that larger vesicles, with less curvature, have homogeneous tight packing under the gel phase because of the lower curvature and require smaller transition enthalpy. On the other hand, for LUV and SUV, with diameters smaller than 500 nm, peak shapes are more expansive. They need considerable energy input to make the phase transition, probably because of the uneven lipid orientations [20]. As shown in Figure 5 and Appendix A, this size-dependent thermal behavior of SOPC vesicles is reasonably consistent with the anisotropy measurement explained before.

Then, we investigated the effect of osmotic pressure on the thermal properties of SOPC vesicles, as shown in Figure 8, which show the DSC peak profiles of GUVs (14,100 nm) and SUVs (100 nm) after repeated freeze-and-thaw cycles (5 to 8 times). GUVs showed a diminution of the trace of the pretransition peak from lamellar gel to ripple gel at around 2 to 3 °C, as shown by arrow (a) and the corresponding increase of peak height for the main transition peak to the Lα phase, shown by arrow (b). In contrast, there was no peak shape change observed for SUVs. These results may indicate that the packing condition of membrane lipids affects the local and temporal phase transition potential.

## 3. Discussion

As explained in the introduction, the mechanism we propose for cytolysis is “a temporal and local phase transfer of membrane lipid from Lα to Mesh_1_ by positive curvature generation” [6,9,11]. Mesh_1_ exists between Lα and V_1,_ which has an undulated lamellar structure with periodical pores (Appendix A). We have confirmed this assumption with the addition of hexa-arginine (R6) to the DOPC/water (60/40) Lα system [11]. This undulated lamellar structure is quite similar to the ripple phase known for the lipid bilayers between the Lα and gel phases, as explained in Section 2.4.2 for the SOPC GUVs. The ripple phase appears because of the periodic local spontaneous curvature in the lipid bilayers, formed due to electrostatic coupling between water molecules and the polar headgroups or the coupling between the membrane curvature and the molecular tilt [20]. Additionally, the cause of ripple formation must correspond to the *SP* change in the Lα structure. It is noteworthy that the existence of the ripple phase was only found for GUVs, which are larger than 1000 nm, enough to avoid curvature-vending energy against cytolysis (Figure 6, Figure 7 and Figure 8). As such, there is a tendency to create periodic undulation for the uniformly spread planar Lα structure with stimuli, namely, for the ripple phase formation by phase transition enthalpy (*ΔH*) and for cytolysis by CPP adsorption.

The cytolysis mechanism we have proposed can be explained by conceptual analogy, visualized by the cartoon shown in Appendix A [6], which is based on the story titled “The man who could walk through walls”, by Marcel Ayme [21]. When the CPP adsorbs to the Lα membrane (Mr. D approaches the wall), Lα turns to Mesh_1_ (the wall surrounding him melts into fluid), and the CPPcan move into the cytosol thorough the temporal Mesh_1_ pore (Mr. D found himself at the other side of the wall); the membrane then returns to Lα (there is a hard wall behind Mr. D). If the membrane (wall) is resistant to changing its curvature, the CPP (Mr. D) will have difficulty penetrating the membrane (walking through the wall). Actually, under the gel state or with small-diameter vesicles, CPPs cannot pass through the membrane, as shown in Appendix A.

In the case where the membrane is flat, the mechanism is reversible for both sides so that CPP concentration would be the same in equilibrium. The systems we are dealing with are cells or vesicles as discrete entities dispersed in the media, with positive membrane curvature. Vesicles have Laplace pressure (*ΔP*) or high vending energy and internalized CPPs face strong resistance from the membranes with negative curvature, as we have proven before [10]. As a result, we could explain the cause of cytolysis suppression by reducing the vesicle size (because of the restriction of lipid mobility); osmotic pressure reduction to enhance positive curvature generation works as long as the membrane is mobile enough to modulate the local structure to Mesh_1_. Taking all the key factors and their effects as a tool, we will further explore how to control CPP cytolysis for the development of a DDS system in combination with the appropriate selection of cargo to be tagged with CPPs.

## 4. Materials and Methods

### 4.1. Materials

Egg yolk phosphatidylcholine (EPC; Tc > −1.5 °C), 1,2-dioleoyl-phosphatidylcholine (DOPC), and 1-stearoyl-2-oleoyl- phosphatidylcholine (SOPC; Tc 6°C) were gifted from NOF Co., Tokyo, Japan, and used as phospholipids that constitute vesicles. As a CPP, arginine octamer (R8) was used. Fluorescein isothiocyanate (FITC), which is a fluorescent label, was attached to R8 via γ-aminobutyric acid (GABA; abbreviated as “FITC–R8”). This compound was prepared by the Futaki laboratory at the Institute for Chemical Research, Kyoto University.

### 4.2. Methods

#### 4.2.1. Preparation of Phosphate-Buffered Saline (PBS(−))

PBS(−) was used as a buffer because the external osmotic pressure of vesicles can be changed by changing its salt concentration. Briefly, 5.0 g of KCl, 5.0 g of KH_2_PO_4_, 72.4 g of Na_2_HPO_4_ 12H_2_O, and 200 g of NaCl were added to a 1 L Erlenmeyer flask, followed by the addition of 800 mL of ultrapure water, and the mixture was stirred at room temperature for one day. Then, the solution was transferred to a 1 L flask, and ultrapure water was added up to 1 L to prepare 25× PBS(−). Then, 1× PBS(−) was prepared by diluting 25× PBS(−) 25 times with ultrapure water. The osmotic pressure of 1× PBS(−) is 280 mOsm.

#### 4.2.2. Preparation of Vesicles

In this study, vesicles with various particle sizes were used. Giant unilamellar vesicles (GUVs) were prepared by using the freeze–thaw dialysis method. Briefly, 50 mg of phospholipids were measured in a glass vial, and CHCl_3_ was added to dissolve it; then, the solvent was evaporated with nitrogen gas to prepare a lipid thin film. The lipid was dried under reduced pressure for 24 h in a vacuum desiccator while being shielded from light. 3 M KCl aqueous solution was added to the lipid thin film, and the lipid concentration was adjusted to 20 mM. The thin film was peeled off from the wall surface by stirring with a vortex mixer. This operation is called the Bangham method, and multilamellar vesicles (MLVs) were obtained. Then, the MLVs were treated for 30 min with a bath-type ultra sonicator to obtain vesicle suspension. The resulted vesicles were unilamellar, with a small diameter. After sealing the vesicles in a plastic tube, the vesicles were rapidly frozen by using liquid nitrogen, followed by thawing at room temperature, stirred with a vortex mixer for 1 min. These operations were repeated six times. Subsequently, the mixture was diluted by 25× PBS(−) to a 5 L beaker and adjusted to 28 mM PBS(−) (56 mOsm). The lipid suspension was transferred to a dialysis tube (Viskase Companies, Inc., Lombard Il. USA; fraction molecular weight: 14,000) and dialyzed for 3 days in the 5 L beaker at room temperature under light-shielding. After dialysis, the lipid suspension was transferred to a light-shielding vial and aged at room temperature for 2 days to prepare the GUV suspension. Vesicles with sizes of 0.1, 0.4, 0.8, 1, and 5 μm were separated by using an extruder.

#### 4.2.3. Phospholipid Concentration Measurement

The concentration of the phospholipids constituting the vesicles differs from the initial concentration because of the dialysis process. Therefore, the final concentration is adjusted by using a concentration measurement kit called c-test Wako. To 1.0 mL of the C-test buffer, 20 μL of a 3.0 g/L BSA standard solution or vesicle suspension was added and incubated at 37 °C for 10 min. After that, the standard solution or the sample’s adsorption was measured by using an ultraviolet–visible spectrophotometer (wavelength: 600 nm). The mol concentration of the phospholipid (Mlipid) is represented by the following Formula (5).
Mlipid (mol/L) = (3.0(g/L) × A_Sa_)/(molecular weight of phospholipid (g/mol) × A_St_)(5)
where A_Sa_ is the adsorption of the GUV, and A_St_ is the adsorption of the standard solution.

#### 4.2.4. Preparation of Trypsin Solution

In the course of membrane permeation, the CPP first adsorbs to the vesicle surface. Therefore, residual CPP adsorbed on the vesicle’s outer surface must be deducted to get the accurate cytolysis (internalization) amount. To do so, trypsin, a basic amino-acid-degrading enzyme, was applied to the test solution, after a penetration test, to decompose the CPP adsorbed on the membrane surface. A method for preparing a trypsin solution is as follows: 5.0 g of trypsin was poured to 200 mL Erlenmeyer flask and 100 mL of 140 mM PBS(−) was added. After stirring the suspension for 2 h in a thermostatic chamber at 37 °C, the suspension was centrifuged (4 °C, 7000× *g*, 20 min) to remove insoluble residue; then, a 5% trypsin solution was prepared. Fractions of 10 mL were collected and stored as trypsin stock solution at −20 °C. For practical use, the trypsin stock solution was diluted 100-fold with PBS(−) to make a 0.05% trypsin solution for the decomposition of CPPs on the membrane surface.

#### 4.2.5. Confirmation of GUV Formation

The prepared vesicles’ size and shape were evaluated by freeze-fractured transmission electron microscope (FF-TEM, JEOL) and differential interference contrast microscope (DICM, Olympus BX53). The GUV solution was sandwiched between glass plates and directly observed in the bright field. For SUVs and LUVs, on the other hand, these frozen solutions were fractured by using a microtome while being maintained at a cryogenic temperature and under high vacuum conditions. Then, the replicas of the fractured surface, deposited by platinum and carbon, were observed by TEM.

#### 4.2.6. Particle Size Distribution Measurement of GUVs

A dynamic light scattering measuring apparatus (DLS, Nicomp 380ZLS, Agilent Technologies, Tokyo, Japan), using an argon laser (532 nm), was used for the particle size distribution measurement of GUVs.

#### 4.2.7. Measurement of Cytolysis Amount of CPPs

The cytolysis amount of CPPs was measured by the following procedures. After adding 28 mM PBS(−) to 600 μL of each prepared vesicle suspension (20 mM), the samples were centrifuged at 10,500 rpm for the 100 nm vesicles and 47,000 rpm for the 400 nm vesicles at 4 °C for 10 min; then, the supernatant was removed. This operation was repeated twice. Osmotic pressure was controlled by adding PBS to each corresponding concentration, as explained in Section 4.2.1. After that, the suspension was kept for 30 min to adjust the vesicles’ curvature to the corresponding osmotic pressure. Phospholipid concentration was adjusted to 10 mM each from the measured lipid concentration, as shown in Section 4.2.3. The 10 mM vesicles allocated to each osmotic pressure were fractionated into four 200 μL portions (to measure with *n* = 4). Then, 20 μL of 100 μM FITC–R8 and 780 μL of various PBS were added so that the total amount would be 1 mL. The molar ratio of lipid/peptide (L/P) was fixed at 1000. After the addition of FITC–R8, incubation was carried out at an appropriate temperature for 10 min. After that, to measure the permeation amount of CPPs separately from the adsorption amount on the membrane surface, a trypsin solution was added and incubated again at 37 °C for 10 min. After the incubation, sample solutions were centrifuged at 10,500 rpm (for the 100 nm vesicles) and at 47,000 rpm (for the 400 nm vesicles) at 4 °C for 10 min to separate the supernatant (decomposed FITC–R8 in the PBS) and the precipitate (vesicles). This operation was done twice. To the precipitate, 160 μL of 10% Triton X-100 and 640 μL of 28 mM PBS(−) were added up to a total volume of 800 μL. This operation disintegrated the vesicles, and the fluorescence measurement (Ex = 495 nm, Em = 520 nm) of internalized FITC–R8 in the vesicles was carried out using a fluorescence spectrophotometer. The obtained fluorescence intensity was converted into concentration by using a calibration plot of FITC–R8. From the fluorescence intensity obtained by fluorescence measurement and the calibration plot previously prepared, the cytolysis amount of FITC–R8 that penetrated the vesicles was calculated. Further, it was converted as the molar ratio of FITC–R8 to lipids using Formula (6).
Amount of permeation(μmol/mol) = ((amount of permeated FITC − R8(g))/(molecular weight of FITC − R8(g/mol)))/((N_lipo_ × N_tot_)/N_A_)(6)
where N_lipo_ is the number of vesicles in 1 mL of suspension, N_tot_ is the number of constituent lipids per one vesicle, and N_A_ is the Avogadro number.

#### 4.2.8. Fluorescence Anisotropy Measurements

Vesicles were labeled with DPH by adding 10 μL of 10 mM DPH/THF solution to 1 mL of liposomal suspension, which was then incubated at 37 °C for 2 h in the dark. Measurements of fluorescence anisotropy were performed with a fluorescence spectrometer (Ex = 351 nm, Em = 430 nm). 

#### 4.2.9. Calorimetric Measurement by DSC

The calorimetric measurements were carried out using MicroCal VP-DSC (Malvern Panalytical Ltd.) in the temperature range of 1–50 °C to obtain the phase transition enthalpy (ΔH). The PBS buffer solution and 2 mM SOPC solutions were used as a reference and a sample, respectively, and all the solutions were degassed under vacuum to eliminate any dissolved air before the measurement. ΔH was calculated from a peak area in the DSC curve using MicroCal Origin software. Both heating and cooling scanning rates were fixed at 45 °C /hour.

## 5. Conclusions

Previously, we confirmed the mechanism for CPP cytolysis as a local and temporal phase transition of membrane lipids from Lα to Mesh_1_ based on the surfactant parameter (*SP*) theory by the adsorption of CPPs, caused by the induction of positive curvature. EPC and SOPC vesicles were used as a cell membrane model in this study. The diameter of the vesicles was changed to investigate the effect of curvature on CPP cytolysis. Against our expectation, vesicle size reduction, which corresponds to a curvature increase, lowered CPP cytolysis. In particular, no cytolysis occurred for 100 nm vesicles, while osmotic pressure reduction, which increases membrane curvature, enhanced CPP cytolysis for the vesicles over 1000 nm. On the other hand, at the gel phase, cytolysis was suppressed regardless of vesicle size. The difference in CPP cytolysis due to the lipid phase was also evident from the confocal microscope observation.

As a result, it is suggested that for CPPs to permeate through the membrane, the size of the vesicles must be over 1000 nm, about the size of living cells, and the membrane must be in a liquid crystal phase. In particular, the phenomenon that the vesicles’ particle size dramatically influences CPP membrane cytolysis is an important discovery.

## Figures and Tables

**Figure 1 ijms-21-07405-f001:**
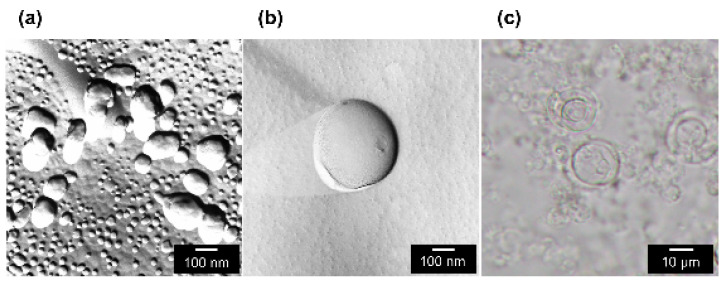
Confirmation of the vesicle formation for 1-stearoyl-2-oleoyl-phosphatidylcholine (SOPC). (**a**) Freeze-fractured transmission electron microscope (FF-TEM) image of small unilamellar vesicle (SUV); (**b**) FF-TEM image of large unilamellar vesicle (LUV); (**c**) optical microscope image of giant unilamellar vesicle (GUV).

**Figure 2 ijms-21-07405-f002:**
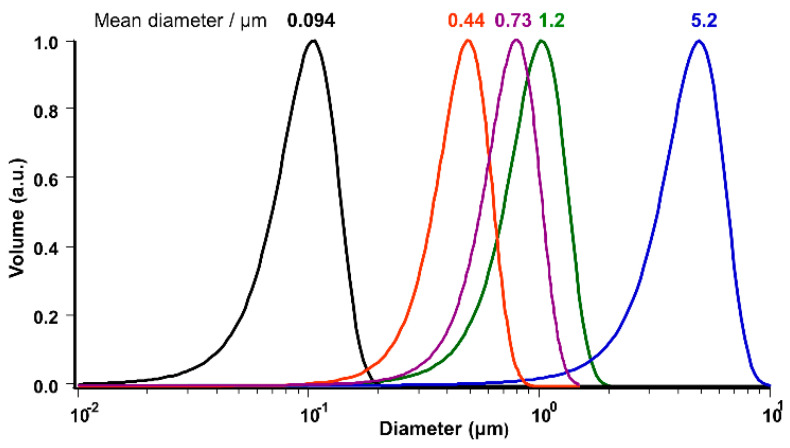
Size distribution analysis for egg yolk phosphatidylcholine (EPC)-vesicles by dynamic light scattering (DLS).

**Figure 3 ijms-21-07405-f003:**
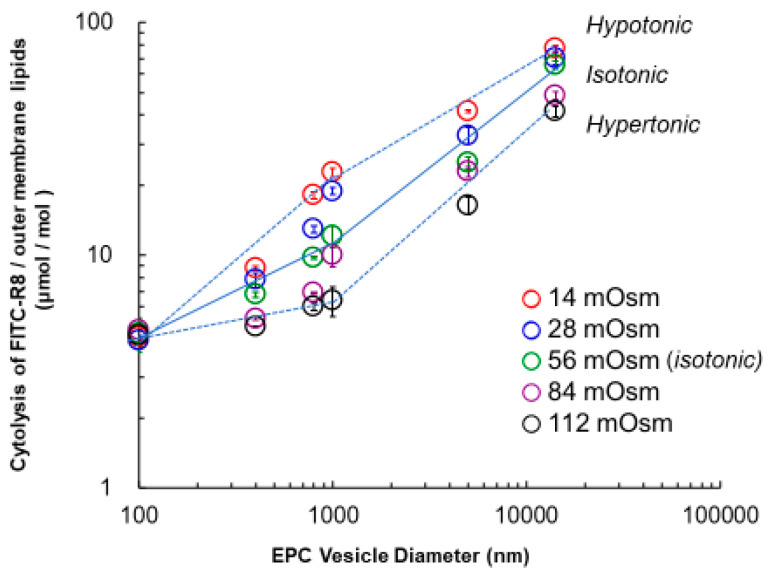
Effect of vesicle size with osmotic pressure change for FITC–R8 cytolysis to EPC vesicles. L/P = 1000, 37 °C, 10 min, *n* = 3; cytolysis amount of FITC–R8/outer membrane lipids (μmol/mol).

**Figure 4 ijms-21-07405-f004:**
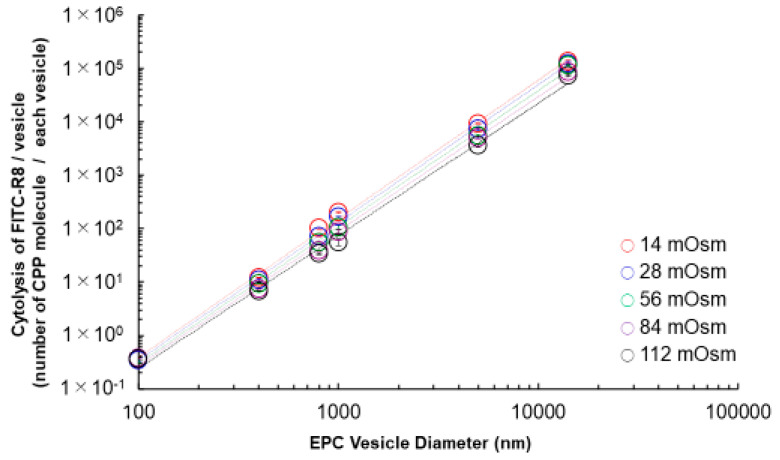
Effect of osmotic pressure for FITC–R8 cytolysis vs. EPC vesicle size. L/P = 1000, 37 °C, 10 min, *n* = 3; cytolysis amount of FITC–R8 (CPP)/vesicle (number of CPP molecule/each vesicle).

**Figure 5 ijms-21-07405-f005:**
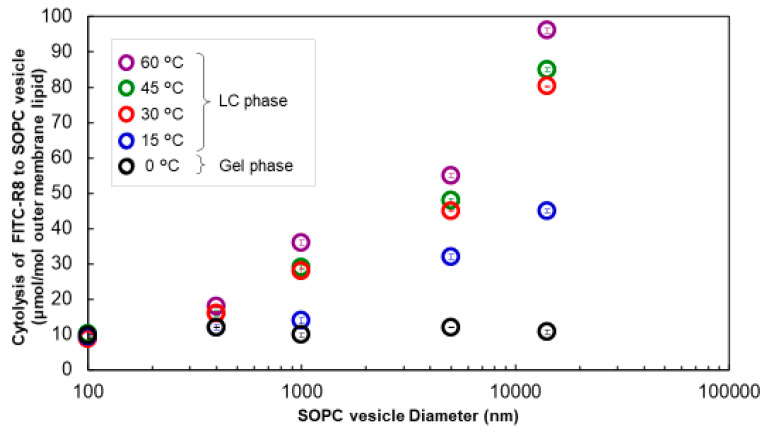
Influence of temperature on cytolysis of FITC–R8 to SOPC vesicle (SOPC; L/P = 1000, 10 min, *n* = 3).

**Figure 6 ijms-21-07405-f006:**
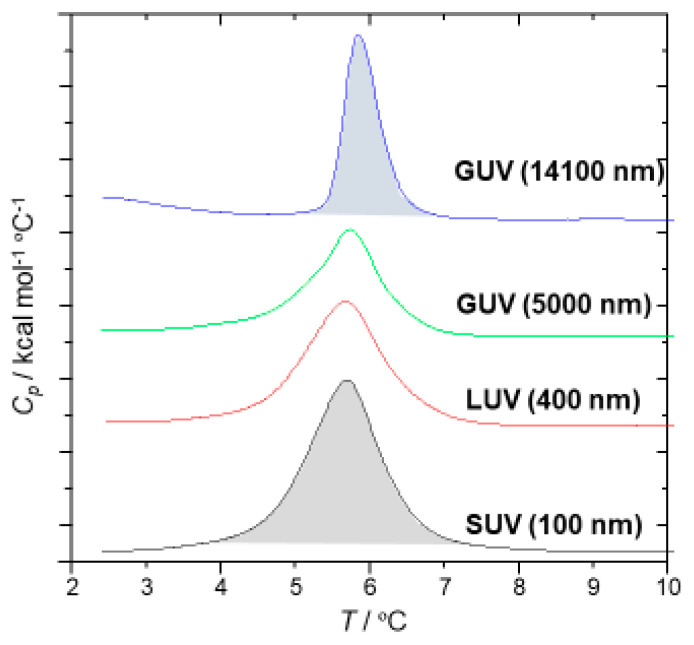
DSC profiles of SOPC vesicles.

**Figure 7 ijms-21-07405-f007:**
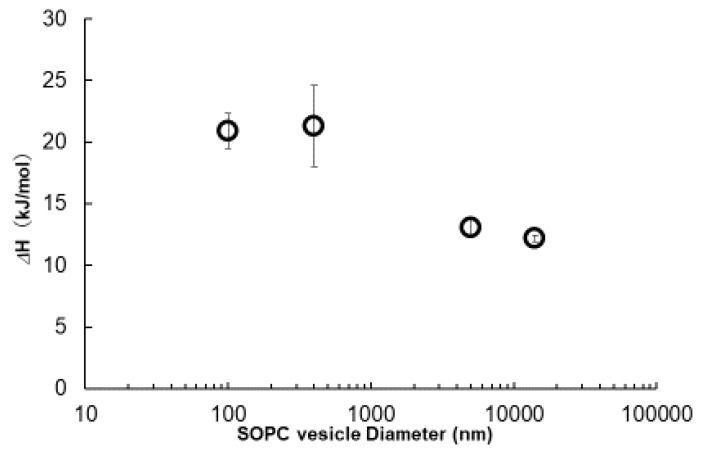
Gel to liquid crystal (LC) phase transition enthalpy (Δ*H*) for SOPC vesicles.

**Figure 8 ijms-21-07405-f008:**
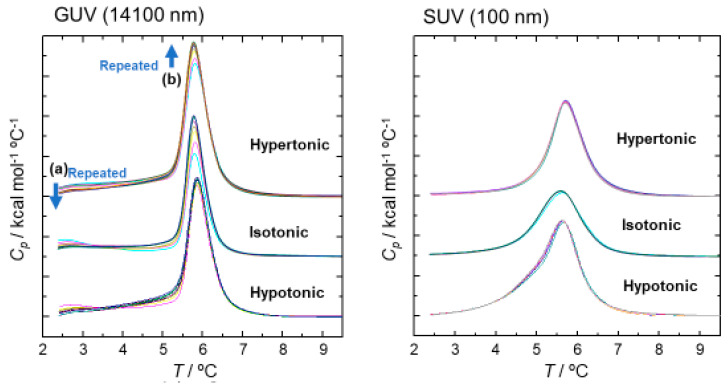
Effect of osmotic pressure on the DSC peak profiles of SOPC vesicles. DSC peaks are recorded for a repeated freeze-and-thaw process (5–8 times). Osmotic pressures (mOsm) are 112 for hypertonic, 56 for isotonic, and 14 for hypotonic. Arrow (a) indicates the diminution of the trace of the ripple phase, and arrow (b) shows the incremental height of the phase transition peak.

**Table 1 ijms-21-07405-t001:** Geometrical parameters of vesicles.

**(a)** EPC.						
	100 nm	400 nm	800 nm	1000 nm	5000 nm	14,100 nm
*N_tot._*, unit	8.06 × 10^4^	1.38 × 10^6^	5.60 × 10^6^	8.77 × 10^6^	2.21 × 10^8^	1.76 × 10^9^
*N_out._*, unit	4.42 × 10^4^	7.08 × 10^5^	2.83 × 10^6^	4.42 × 10^6^	1.11 × 10^8^	8.80 × 10^8^
*N_int._*, unit	3.63 × 10^4^	6.75 × 10^5^	2.77 × 10^6^	4.34 × 10^6^	1.10 × 10^8^	8.79 × 10^8^
*N_lipo_*, unit/mL	1.49 × 10^13^	8.71 × 10^11^	2.15 × 10^11^	1.37 × 10^11^	5.45 × 10^9^	5.82 × 10^8^
**(b)** SOPC.						
	100 nm	400 nm	800 nm	1000 nm	5000 nm	14,100 nm
*N_tot._*, unit	6.81 × 10^4^	1.17 × 10^6^	4.73 × 10^6^	7.41 × 10^6^	1.87 × 10^8^	1.49 × 10^8^
*N_out._*, unit	3.74 × 10^4^	5.98 × 10^5^	2.39 × 10^6^	3.74 × 10^6^	9.35 × 10^7^	7.44 × 10^8^
*N_int._*, unit	3.07 × 10^4^	5.71 × 10^5^	2.34 × 10^6^	3.67 × 10^6^	9.31 × 10^7^	7.43 × 10^8^
*N_lipo_*, unit/mL	1.50 × 10^13^	8.75 × 10^11^	2.16 × 10^11^	1.38 × 10^11^	5.48 × 10^9^	6.89 × 10^8^

*N_tot._*: Number of lipids per one vesicle; *N_out_*_._: Number of outer lipids per one vesicle; *N_int._*: Number of internal lipids per one vesicle; *N_lipo_*: Number of vesicles per 1 mL of liposomal suspension.

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
