# Peer review of "Effect of Vesicle Size on the Cytolysis of Cell-Penetrating Peptides (CPPs)"

_ijms, 2020, doi:10.3390/ijms21197405_

Round 1

Reviewer 1 Report

The authors study the effect of the particle size of phospholipid vesicles on cytosis of cell-penetrating peptides (CPP) with the main aim to find out the critical factor(s) affecting cytolysis. The presented results show that a decrease in the vesicle diameter reduces the cytolysis until suppression. This finding is attributed to a restriction of membrane lipid mobility. The effect of osmotic pressure in relation with the vesicle size is also investigated. 

I had difficulties in revising this work due to the lack of clarity in the presentation and interpretation of the results. Notably, the English has to be improved because it may lead to misunderstandings.

Please find bellow a couple of suggestions but the manuscript requires a deep revision in order to make the message clear.

Lines 36-37: “We have revealed that CPP adsorbed on the lipid membrane changes its curvature by changing the surfactant parameter (SP) [5-7].” Could the authors put this in the context of the research of the field and mention if other related works have been published? It’s not really clear from the introduction which was the first work on the surfactant parameter measurements in relation with lipid membrane changes induced by CPP adsorption.

Lines 13 & 38: DDS not explained

Line 16: What CPC means? Should it be CPP instead of CPC? Idem line 67.

Line 45: V1 and H1 phase are not explained

Line 49: Could the authors shortly explain the L\alpha and Mesh1phases?

Lines 54-56: There should be only one sentence. Moreover, I would make a distinction between erythrocytes, dermal fibroblasts and keratinocytes, on one side, which are types of cells and GUVs, on the other side, which are model systems of cells.

Lines 58-59: Please revise seethe sentence: “As a result, we found that adsorption and cytolysis assumed to occur successively by CPP molecule come to contact with membrane lipid.”

Line 64: “report.The …” extra space missing

Line 65: It’s not fully clear what the authors mean here by particle size. Please be more precise! Do you mean vesicles’ size?

Line 77: Please clarify “stimuli effecting to produce SP”.

Line 81: “The formation … was …” instead of “were”. Please explain FF-TEM and DICM acronyms.

Line 84: SOPC not introduced here but only in section 2.3.2.

Line 86: please explain what EPC means. Could the authors comment on the size distribution obtained for these vesicles? The same size distribution could be provided for SOPC vesicles as well?

Line 90: Small case for “geometrical”

Line 92: “Supplementary Figure3” extra space missing

Line 101: “… the effect … was” instead of “were”

Lines 108: “previous studies”: please indicate these references

Line 113: please indicate what L/P ratio means.

Line 117: “There are … relationships” instead of “relationship” The meaning of the full sentence is not fully clear. Please revise. In fact, the whole paragraph needs to be re-written because the message is really not clear. Could you also please give a reference for the bimodal linear relationships between cytolysis and vesicle size?

Lines 199: “There are inverse relationships…” instead of relationship.

Line 202: “calorimetry” instead of “colorimetry”

Line 205 Use “were almost the same”

Overall, for consistency sake, please check the guidelines of the journal for the instructions of the references: are they before or after the “.”? With or without an extra space? Idem for Figures call: please decide if you use Figure of Fig.

A couple of references contain errors in their titles.

Author Response

 Thank you very much for your kind comments and suggestions. We are very much encouraged and inspired to make a point of this article clearer to the readers.

We made a major revision for the Introduction to correspond to the reviewer's suggestion and made necessary revisions to the points as listed below.

 Please kindly review the revised text and the following responses.

Lines 36-37: “We have revealed that CPP adsorbed on the lipid membrane changes its curvature by changing the surfactant parameter (SP) [5-7].” Could the authors put this in the context of the research of the field and mention if other related works have been published? It’s not really clear from the introduction which was the first work on the surfactant parameter measurements in relation with lipid membrane changes induced by CPP adsorption.

(KS) We made significant revisions for Introduction to correspond to this comment. Please refer to Lines 36-51 of the revised text. We explained why we took a physicochemical approached to investigate CPC cytolysis in view of the structural modification of lipid bilayers as self-assembled liquid crystal. As explained in the revised text, SP is an established geometrical surfactant parameter that is ruling the structure of liquid crystals (LC). By utilizing this SP theory for the LC structure, we have shown that CPC adsorption to the membrane lipid generates positive curvature (SP change) to induce phase transition and structural change for LC of membrane lipids.  

Lines 13 & 38: DDS not explained

(KS) In Lines 13 & 34 of the revised text, we explained DDS as a "drug delivery system," and DDS is used at Line 274.

Line 16: What CPC means? Should it be CPP instead of CPC? Idem line 67.

(KS) These are topological mistakes and revised to CPP at Line 16 and 81.

Line 45: V1 and H1 phase are not explained

(KS) In Lines 44 of the revised text, the meaning of V1 and H1 are explained as specified types of LC structures, which geometries are shown in Supplementary Figure S1.

Line 49: Could the authors shortly explain the Lα and Mesh1phases?

(KS) Same as explained for V1 and H1, La is explained at Line 42 as a hydrated lamellar liquid crystal (Lα), and Mesh1 is explained at Line 50-51 as “Mesh1 has a unique structure of undulated lamellar bilayers with periodical pores to connects water layers”.  All these geometries are shown in Figs. S1 and S2.

Lines 54-56: There should be only one sentence. Moreover, I would make a distinction between erythrocytes, dermal fibroblasts and keratinocytes, on one side, which are types of cells and GUVs, on the other side, which are model systems of cells.

(KS) In Line 52-53 of the revised text, we explained how the idea of physicochemical analysis of CPP cytolysis came up to us. And in Line 54-61, we explained R8 and R6, which are CPP, independently changed LC structure by increasing curvature (changing SP) from La to eventually H1 through Mesh1 and V1.

We first used erythrocyte, as an appropriate cell to prove our hypothesis.

Because of its negative mean curvature, erythrocyte was resistant to cytolysis. As explained in Line 65-70, we confirmed our hypothesis of cytolysis mechanism by using erythrocyte. Also, we could control cytolysis by membrane curvature change with a different osmotic pressure of medium. Normal cells such as fibroblasts and keratinocytes are with positive membrane curvature. So that cytolysis level is high., and it is impossible to enhance positive curvature further by low osmotic pressure to keep cell viability. (instead of osmotic pressure, we have confirmed curvature control of cytolysis by Hoffmeister effect of solute  as shown in ref 5 and 13)  GUV is used as a model cell without any biological functions, and size is about the same to normal cells.    

Lines 58-59: Please revise seethe sentence: “As a result, we found that adsorption and cytolysis assumed to occur successively by CPP molecule come to contact with membrane lipid.”

(KS) At Line 70-71 of the revised text, we changed to "As a result, we found that adsorption and cytolysis were assumed to occur successively when CPP molecule came to contact with membrane lipid.”

Line 64: “report.The …” extra space missing

(KS) Made as a new paragraph at Line 77 to 78 of the revised text.

Line 65: It’s not fully clear what the authors mean here by particle size. Please be more precise! Do you mean vesicles’ size?

(KS) All the "particle size" in the text was changed to "vesicle size"  We changed the Title to "Effect of Vesicle Size on the Cytolysis of Cell-Penetrating Peptide (CPP)."

Line 77: Please clarify “stimuli effecting to produce SP”.

(KS) We changed the sentence by replacing ”to the stimuli effecting to reduce SP value” to “to the reduced osmotic pressure leading smaller SP value” at Line 91-92 of the revised text.

Line 81: “The formation … was …” instead of “were”. Please explain FF-TEM and DICM acronyms.

(KS) “were” was replace to “was” and “FF-TEM” and “DICM” are explained as “freeze-fractured transmission electron microscope (FF-TEM)” and “differential interference contrast microscope (DICM)" at Line 96-97 of the revised text.

Line 84: SOPC not introduced here but only in section 2.3.2.

(KS) “1-stearoyl-2-oleoyl-phosphatidylcholine (SOPC)" is inserted at Line 95 of the revised text, which comes before Figure 1.

Line 86: please explain what EPC means. Could the authors comment on the size distribution obtained for these vesicles? The same size distribution could be provided for SOPC vesicles as well?

(KS) “Egg yolk phosphatidylcholine (EPC)" was used to Line102 of the revised text. Figure 2 shows the size distribution of EPC by DLS measurement, which corresponds to the sizes shown in Table 1 (a) in the round number. SOPC has the same distribution, as shown in Table 1 (b) (DLS data are not shown).

Line 90: Small case for “geometrical”

(KS) Changed to “geometrical” at Line 110 of the revised text.

Line 92: “Supplementary Figure3” extra space missing

(KS) Changed to “Supplementary Figure S3” at Line112-113 of revised text.

Line 101: “… the effect … was” instead of “were”

(KS) Changed to "was" at  Line 121 of the revised text.

Lines 108: “previous studies”: please indicate these references

(KS) References are added at Line 130 of the revised text.

Line 113: please indicate what L/P ratio means.

(KS) “molar ratio of lipid/CPP peptide (L/P)" is added at Line 123 of the revised text, which comes before Figure 3 caption.

Line 117: “There are … relationships” instead of “relationship” The meaning of the full sentence is not fully clear. Please revise. In fact, the whole paragraph needs to be re-written because the message is really not clear. Could you also please give a reference for the bimodal linear relationships between cytolysis and vesicle size?

(KS) The paragraph is revised, as shown in Line 135-141 of the revised text.

Lines 199: “There are inverse relationships…” instead of relationship.

(KS) Corrected to “There are inverse relationships…” at Line 207 of revised text.

Line 202: “calorimetry” instead of “colorimetry”

(KS) Corrected to “calorimetry” at Line 209 of the revised text.

Line 205 Use “were almost the same”

(KS) Changed to “were almost the same” at Line 212 of the revised text.

Overall, for consistency sake, please check the guidelines of the journal for the instructions of the references: are they before or after the “.”? With or without an extra space? Idem for Figures call: please decide if you use Figure of Fig.

A couple of references contain errors in their titles.

(KS) Thank you for the comments. All points are checked for the entire text.

We very much appreciate your comments in details and hope the revisions mentioned above will fulfill your suggestion.  

Reviewer 2 Report

Sakamoto et al. studied the cytolysis mechanism of a cell-penetrating peptide by varying the size of model vesicles and osmotic pressure. Studies were adequately designed and experiments were thoroughly conducted. The findings are interesting that the lipid phase state is an important factor for cytolysis to occur in addition to the membrane curvature. I recommend publishing this work after minor revision (text editing).

Minor comments:

DDS acronym not defined. Please thoroughly check for grammar and spelling.

Author Response

Thank you very much for your kind comments and suggestions. We are very much encouraged and inspired to make a point of this article clearer to the readers.

We made a major revision for the Introduction to correspond to the reviewer's suggestion.

Please kindly review the revised text.

Kazutami Sakamoto